# Advancing Personalized Learning with Neural Collapse for Long-Tail Challenge

**Hanglei Hu** [* 1]   **Yingying Guo** [* 1]   **Zhikang Chen** [2]   **Sen Cui** [2]   **Fei Wu** [3]   **Kun Kuang** [3]   **Min Zhang** [† ‡ 1]   **Bo Jiang** [† 1]

## Abstract

Personalized learning, especially data-based methods, has garnered widespread attention in recent years, aiming to meet individual student needs. However, many works rely on the implicit assumption that benchmarks are high-quality and well-annotated, which limits their practical applicability. In real-world scenarios, these benchmarks often exhibit long-tail distributions, significantly impacting model performance. To address this challenge, we propose a novel method called **N**eural-**C**ollapse-**A**dvanced personalized **L**earning (NCAL), designed to learn features that conform to the same simplex equiangular tight frame (ETF) structure. NCAL introduces text-modality collapse (TC) regularization to optimize the distribution of text embeddings within the large language model (LLM) representation space. Notably, NCAL is model-agnostic, making it compatible with various architectures and approaches, thereby ensuring broad applicability. Extensive experiments demonstrate that NCAL effectively enhances existing works, achieving new state-of-the-art performance. Additionally, NCAL mitigates class imbalance, significantly improving the model's generalization ability. Code is available at `https://github.com/llm4edu/NCAL_ICML2025.git`.

## 1. Introduction

Personalized learning is an important downstream task in intelligent education, aiming to address the unique needs, preferences, and abilities of individual learners (Shemshack & Spector, 2020; Bernacki et al., 2021; Wang et al., 2024a). It primarily encompasses two paradigms: rule-based and data-based methods. Rule-based methods (Skinner, 1961;

---

[*]Equal contribution[†] Corresponding author[‡]Project leader [1]East China Normal University [2]Tsinghua University [3]Zhejiang University. Correspondence to: Min Zhang <mzhang@cs.ecnu.edu.cn>, Bo Jiang <bjiang@deit.ecnu.edu.cn>.

*Proceedings of the 42nd International Conference on Machine Learning*, Vancouver, Canada. PMLR 267, 2025. Copyright 2025 by the author(s).

Anderson et al., 1995; Raj & Renumol, 2019; Pukkhem & Vatanawood, 2011) rely on predefined rules and expert knowledge to guide personalization, while data-based methods (Ayeni et al., 2024; Tapalova & Zhiyenbayeva, 2022; Conati et al., 2021; Wang et al., 2024b; Han et al., 2024) leverage large-scale data and advanced machine learning techniques to dynamically adapt and optimize the learning experience. Due to their flexibility and efficiency in meeting individual needs, data-driven methods have gradually become the mainstream approach. Therefore, in this paper, we focus on **data-based personalized learning**.

Data-based personalized learning is widely applied in important educational scenarios such as knowledge tracing (Ghosh et al., 2020; Sun et al., 2025), cognitive diagnosis (Wang et al., 2020; 2024b), computer adaptive testing (Zhuang et al., 2023; Mousavinasab et al., 2021) and so on. In recent years, researchers have progressively advanced toward deeper exploration, not only leveraging more complex architectures to enhance the expressive power of models (Lu et al., 2021; Grawemeyer et al., 2016) but also focusing on constructing larger-scale and more diverse datasets (Choi et al., 2024; Sun et al., 2019; Huang et al., 2016) to improve the generalization ability of models across different contexts. For example, in cognitive diagnosis tasks, by thoroughly analyzing students' learning behaviors and cognitive characteristics, personalized diagnostic reports can be generated to help teachers monitor students' progress and identify learning bottlenecks in real time, thus providing technical support for student-centered teaching models (Wang et al., 2020; Lu et al., 2021). However, these methods are often based on an implicit assumption that the data used in data-driven approaches is of high quality and well-annotated.

In real-world scenarios, the collection of high-quality and standardized educational data is both challenging and impractical due to concerns over protecting the personal privacy of minors and the reliance on annotations from educational experts (Dominguez et al., 2010; Enders et al., 2008). As a result, benchmarks for data-based personalized learning naturally exhibit imbalance (or long-tail) distributions (Menon et al., 2020; Kanchon et al., 2024; Zhang et al., 2024). However, existing research has largely overlooked this problem. Experiments shown in Figure 1 (a) and (d) demonstrate that the performance of the Qwen2.5 varies with changes in data balance. When the $\tau = 0.25$ value is

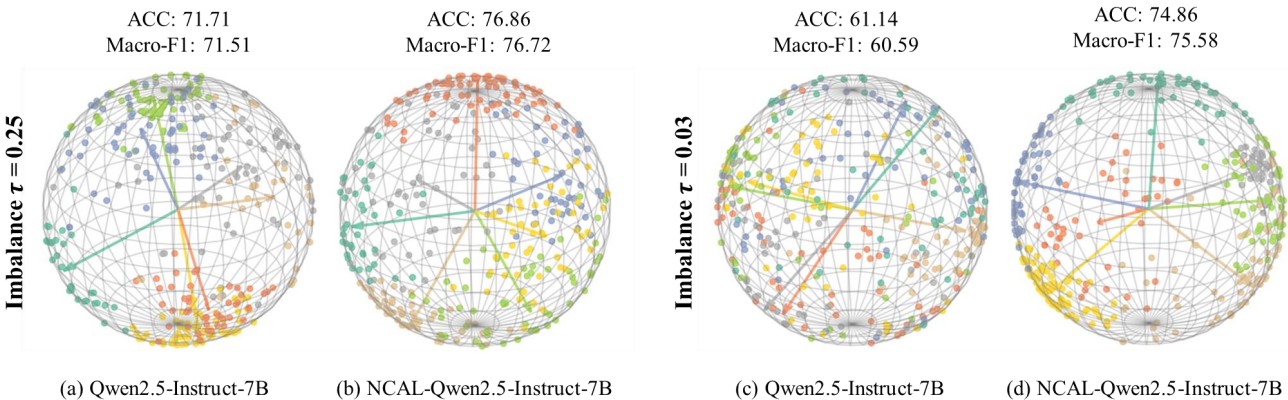

Figure 1: Neural collapse degree visualization in Qwen2.5 (Qwen et al., 2025) on TMWPL. Arrows represent the directions of category centers, while points indicate sample features, with colors corresponding to categories. (a) and (b) show the final classification for the baseline and our NCAL under the $\tau = 0.25$ setting. (c) and (d) illustrate the experimental results in the $\tau = 0.03$. NCAL clearly strengthens the ETF structure, leading to improve generalization performance.

higher, indicating weaker data imbalance, the model performs better. Conversely, as the $\tau = 0.03$ value decreases and data imbalance worsens, the model's performance deteriorates, dropping from 71.71 (Figure 1 (a)) to 61.14 (Figure 1 (c)). Additionally, the distribution of class centers transitions from being dispersed to becoming increasingly aggregated. **This naturally raises a pivotal question:**

> *How can the data imbalance or long-tail distributions in data-based personalized learning be studied and explored to enhance the robustness of intelligent education models?*

The recently discovered neural collapse (NC) phenomenon has emerged as a groundbreaking insight into the optimal structure of visual representations in the field of computer vision (Papyan et al., 2020; Li et al., 2023; Zhang et al., 2022). This phenomenon reveals that when a model achieves zero training error on a sufficiently large and balanced dataset, the last-layer features within the same class converge to their class center. Furthermore, these class centers, along with their corresponding classifier vectors, align with the vertices of a simplex equiangular tight frame (ETF). A simplex ETF describes a geometric structure where vectors maximize pairwise angles while maintaining equal norms. Inspired by this phenomenon, we aim to investigate the geometric structures of representations during the optimization in data-based personalized learning methods, seeking to enhance their theoretical foundation and practical performance.

In this paper, we propose a novel method called **N**eural-**C**ollapse-**A**dvanced personalized **L**earning (NCAL), designed to learn features that conform to the **equiangular tight frame (ETF)** structure. Specifically, NCAL introduces **text-modality collapse (TC)** regularization to optimize the distribution of text embeddings within the large language model (LLM) representation space, thereby enhancing the robustness and generalization ability of LLM

LoRA fine-tuned models under long-tail data distributions. Through theories and experiments, we demonstrate that integrating NC into personalized learning leads to significant improvements across multiple evaluation metrics. Our contributions are mainly summarized as follows:

- To the best of our knowledge, we are the first paper to extend the neural collapse phenomenon to data-based personalized learning and to explore its underlying representational mechanism for generalization capability.

- We further investigate the impact of class imbalance on the performance of LLM-based methods in personalized learning, filling a critical gap in existing research. We propose the NCAL to improve the generalization of LoRA fine-tuned LLMs in class-imbalanced settings.

- Through theoretical analysis and extensive experiments, we validate the effectiveness of combining NCAL with existing LoRA fine-tuning methods for LLMs and demonstrate its significant improvement.

## 2. Related Work

**Personalized learning.** Personalized learning emphasizes customizing educational experiences to address the unique needs, preferences, and abilities of individual learners (Shemshack & Spector, 2020; Bernacki et al., 2021). Over the years, it has evolved into two primary paradigms: rule-based and data-based methods. Rule-based methods (Skinner, 1961; Anderson et al., 1995; Raj & Renumol, 2019; Pukkhem & Vatanawood, 2011) rely on predefined rules and expert knowledge to guide personalization. In contrast, data-based methods (Ayeni et al., 2024; Graesser et al., 2004; Grawemeyer et al., 2016; Tapalova & Zhiyenbayeva, 2022; Conati et al., 2021) harness insights from large-scale

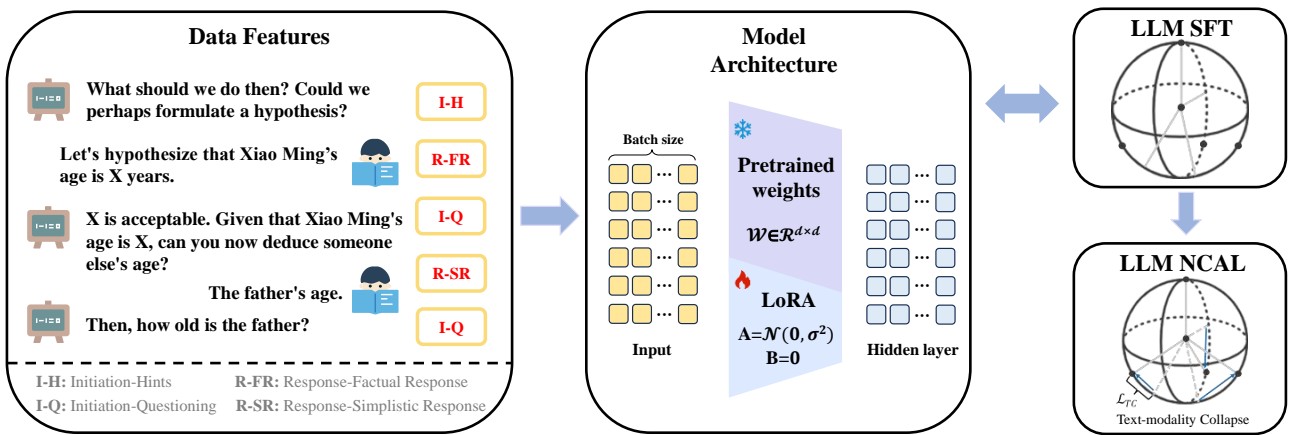

Figure 2: The architecture of the proposed NCAL. While traditional LLM fine-tuning methods (*e.g.*, LoRA) fail to achieve ETF in their final text representations, our NCAL introduces the Text-modality Collapse (TC) loss to align text representations with a simple ETF structure, enabling robust classification performance on long-tail distributions.

data, frequently employing advanced machine learning techniques, to dynamically adapt and optimize learning experiences. Rule-based methods often require significant investments of time and resources, making data-based methods the dominant paradigm in personalized learning. In this paper, we focus primarily on data-based personalized learning.

**Data-based personalized learning.** The ultimate goal of data-based personalized learning is to have a learning system (Shemshack & Spector, 2020) that can self-adjust according to learners' characteristics and needs. Based on the rule-based foundation, data-based personalized learning is gradually progressing towards this goal. Recent studies have not only used diverse techniques to enhance the representational and analytical capabilities of models (Zhou et al., 2020; 2016; Bulathwela et al., 2020; Graesser et al., 2004), but also the assessments of students' ability and cognitive level are refined and automated (Askarbekuly & Aničić, 2024; Cohn et al., 2024; Chang & Ginter, 2024), which construct students' portraits better. For example, in the personalized recommendation task, the browsing logs of different students are analyzed to understand learners' background knowledge and cognitive state, and then recommend personalized content that is novel and in the nearest development zone (Bulathwela et al., 2020). However, these methods are often based on an implicit assumption that the data used in data-driven approaches is of high quality and well-annotated. Therefore, in this paper, we propose a novel method to address the challenge in the real-world setting.

**Neural collapse.** Neural collapse is a geometric phenomenon where the structured arrangement of features enhances the separation between categories. This phenomenon was first discovered by Papyan et al. (Papyan et al., 2020) during training on balanced datasets, enabling classifiers to make more stable decisions as training progresses. In recent years, research on neural collapse has expanded into various

fields, including imbalanced learning (Liu et al., 2023; Zhu et al., 2021; Tirer & Bruna, 2022; Kothapalli, 2022), contrastive continual learning (Montmaur et al., 2024), transfer learning (Galanti et al., 2021), federated learning (Li et al., 2023; Huang et al., 2023) and so on. Although neural collapse has been validated in various domains, existing research has largely been limited to linear classifiers. In this paper, we are the first to apply neural collapse to address the long-tail problem in data-based personalized learning and improve the robustness of large language models during LoRA fine-tuning in personalized learning scenarios.

## 3. Preliminary

In this section, we first give a detailed description of LoRA-based fine-tuning for LLMs in Section 3.1, and then introduce the problem definition of neural collapse in Section 3.2.

### 3.1. Data-Based Personalized Learning

Data-based personalized learning emphasizes the processing and analysis of students' learning and behavioral data. Owing to LLMs' exceptional ability to handle complex language tasks, their application in data-based personalized learning has become a prominent trend (Liu et al.; Shridhar et al., 2022; Cheng et al., 2022; Zhang et al., 2023). Modern LLMs typically adopt the Transformer architecture, which processes sequential data through multiple layers of self-attention and feed-forward networks. The fundamental computation in each transformer layer can be expressed as:

$$h_l = \text{FFN}(\text{LayerNorm}(h_{l-1} + \text{MultiHead}(h_{l-1}))), \quad (1)$$

where $h_l$ represents the hidden state at layer $l$, FFN means the feed-forward network, LayerNorm denotes the layer normalization and the multi-head attention mechanism (Multi-

Head) is computed through $H$ parallel attention heads:

$$\text{MultiHead}(X) = \text{Concat}(\text{head}_1, ..., \text{head}_H)W^O, \quad (2)$$

$$\text{head}_i = \text{Attention}(XW_i^Q, XW_i^K, XW_i^V). \quad (3)$$

For text encoding, LLMs first transform input tokens into continuous representations through an embedding layer:

$$z = E(x) + P, \quad z \in \mathbb{R}^{L \times D} \quad (4)$$

where $x$ represents the input token sequence, $E$ is the token embedding matrix, $P$ is the positional encoding, $L$ is the sequence length, and $D$ is the model dimension.

Consequently, an increasing body of research in personalized learning is focusing on how to effectively fine-tune large language models to address the unique needs of individual learners, harnessing the models' robust representational capabilities to enhance learning outcomes.

Of all the fine-tuning methods, LoRA aims at fine-tuning LLMs on specific tasks, reducing the number of trainable parameters and minimizing performance loss (Hu et al., 2021). Inspired by (Aghajanyan et al., 2020), LoRA assumes that weight updates during adaptation also have a low "intrinsic rank", and constrains the update of the pre-trained weight matrix $W_0 \in \mathbb{R}^{d \times k}$ through a low-rank decomposition $\Delta W = BA$, where $B \in \mathbb{R}^{d \times r}$ and $A \in \mathbb{R}^{r \times k}$ include trainable parameters. For these matrixes, $d$ denotes the rank of LLMs, $r$ is the rank of lora module, and $k$ is the output dimension of the original weight matrix.

To enhance parameter efficiency and reduce memory usage, LoRA modifies the forward pass as follows:

$$y = W_0 x + \Delta W x = W_0 x + \frac{\alpha}{r} BAx, \quad (5)$$

where $x \in \mathbb{R}^k$ denotes the input, $y \in \mathbb{R}^d$ represents the output after a low-rank adjustment and the rank $r \ll \min(d, k)$ because the parameter efficiency of LoRA must be preserved during inference. $\alpha$ is a constant parameter for scaling. The matrix $A$ is initialized as a Gaussian distribution with a mean of and the matrix $B$ is initialized as a zero matrix.

### 3.2. Neural Collapse

Neural collapse refers to a phenomenon observed in the final stage of training when the model achieves zero training error and datasets are balanced. At this stage, the last layer of the feature and structure of the classifier together form a simplex equiangular tight frame (ETF), which is defined as:

**Definition** (Simplex equiangular tight frame). A matrix consisting of $N$ vectors $m_i \in \mathbb{R}^v$ with $v \geq N - 1$ meaning the dimension of the feature space, may form a simplex ETF if it satisfies the following formula:

$$M = \sqrt{\frac{N}{N-1}} U(\mathbf{I}_N - \frac{1}{N} \mathbf{1}_N \mathbf{1}_N^T), \quad (6)$$

where $M = [\mathbf{m}_1, \cdots, \mathbf{m}_N] \in \mathbb{R}^{v \times N}$, $\mathbf{U} \in \mathbb{R}^{v \times N}$ satisfies a rotation, $N$ is the number of classes and $\mathbf{U}^T \mathbf{U} = \boldsymbol{I}_N$, $\boldsymbol{I}_N$ is the identity matrix, $\mathbf{1}_N \in \mathbb{R}^{N \times 1}$ is an all-ones vector. Vectors in a simplex ETF have an equal $\ell_2$ norm and consistent pairwise angles, as captured in the following formula:

$$\mathbf{v}_i^T \mathbf{v}_j = \frac{N}{N-1} \delta_{i,j} - \frac{1}{N-1}, \forall i, j \in [1, N], \quad (7)$$

where $\delta_{i,j}$ is the Kronecker delta function, which equals 1 when $i = j$ and 0 otherwise. This ensures the maximal equiangular separation is pairwise angle $-\frac{1}{N-1}$.

The neural collapse phenomenon (Papyan et al., 2020) is guaranteed by the following three interrelated characteristics: features collapse to the class prototypes (**NC1**), prototypes collapse to simplex ETF (**NC2**), and classifiers collapse to the same simplex ETF (**NC3**).

**NC1:** As training advances, the variation within each class in the final layer will collapse, *i.e.*, for all $n$, $\sum_W^n \to 0$. $\Sigma_W^n = \frac{1}{j_n} \sum_{i=1}^{j_n} (\mathbf{u}_{n,i} - \mathbf{u}_n)(\mathbf{u}_{n,i} - \mathbf{u}_n)^T$, where $\mathbf{u}_{n,i}$ denotes the features obtained from the i-th sample of class $n$, and $\mathbf{u}_n$ is the class $n$'s prototype.

**NC2:** All classes converge to the vertices of a simplex ETF with identical pairwise angles, *i.e.*, $\hat{\mathbf{u}}_n = (\mathbf{u}_n - \mathbf{u}_G) / \|\mathbf{u}_n - \mathbf{u}_G\|$, as defined in Equation (7). $u_G = \sum_{n=1}^{N} \sum_{i=1}^{J_n} \mathbf{u}_{n,i}$ denotes the global mean of the features.

**NC3:** The vectors in the final layer of the linear classifier matrix converge to a simplex ETF as prototypes, *i.e.*, $\tilde{\mathbf{u}}_n = \mathbf{m}_n / \|\mathbf{m}_n\|$, where $u_n$ is the classifier weights of class $n$.

## 4. Methodology

In this section, we present our NCAL approach to address the long-tail problem in educational data. We first extend the concept of NC to text-based LLMs in Section 4.1, introducing the TCD. In Section 4.2, we describe our TC regularization, which integrates an NC-inspired loss with LoRA fine-tuning. This approach aims to optimize LLM representations towards a simplex ETF structure. Finally, in Section 4.3, we provide a theoretical analysis of our method.

### 4.1. Neural Collapse in Text-Based LLMs

While neural collapse (NC) has been extensively studied in computer vision and other domains, its application to pure text-based LLMs remains relatively unexplored. We propose a theoretical framework to extend NC concepts to text-based LLMs, focusing on the structure of text representations in the embedding space.

Let $h_l(\cdot)$ be the text encoder at layer $l$. We define the text-modality collapse degree (TCD) for text representations:

$$\Delta_{TCD} = \text{Avg}_{i \neq j} \{\langle h_l(t_i), h_l(t_j) \rangle - T_W \cdot \mu\}, \quad (8)$$

where $\mu = -\frac{1}{N-1}$, $t_i$ and $t_j$ are text inputs, $N$ is the number of classes, and $T_W$ is a fixed-length constraint for text representations. The TCD measures the degree of separation among text representations, reflecting their proximity to a simplex equiangular tight frame (ETF) structure. A lower $\Delta_{TCD}$ indicates that text representations are closer to forming a simplex ETF, which is a desirable property for classification tasks. From Figure 2, we hypothesize that:

(1) A higher degree of neural collapse (lower $\Delta_{TCD}$) leads to stronger generalizability in text classification tasks.

(2) The presence of data imbalance during the training process adversely impacts the degree of the neural collapse phenomenon, thereby compromising the model's generalization performance capabilities.

These hypotheses can be formally expressed as:

$$\tau \downarrow \Rightarrow \Delta_{TCD} \uparrow \Rightarrow \text{Acc} \downarrow, \qquad (9)$$

where the ratio $\tau = \frac{\text{counts of the least classes}}{\text{counts of the most classes}}$ serves as a quantitative metric characterizing the severity of class distribution imbalance within the dataset. Based on these theoretical insights, we design our method to optimize the LLM's representations towards a simplex ETF structure, even under class imbalance conditions.

### 4.2. Text-modality Collapse Regularization

We introduce the text-modality collapse (TC) loss and integrate it with LoRA fine-tuning. The TC loss is defined:

$$\mathcal{L}_{\text{TC}} = \sum_{1 \le i < j \le B} \mathbb{I}(y_i \neq y_j) \left( \langle h_l(t_i), h_l(t_j) \rangle - T_W \cdot \mu \right)^2, \quad (10)$$

subject to:

$$\|h_l(t_i)\|_2 \le \sqrt{T_W}, \forall i \in \{1, ..., B\}, \qquad (11)$$

where $B$ is the batch size, $y_i$ is the class of the $i$-th sample, and $\mathbb{I}(y_i \neq y_j)$ is an indicator function.

For implementation, we utilize LoRA to efficiently adapt the LLM weights $\mathbf{W}_0$ using trainable rank-decomposition matrices $\mathbf{B} \in \mathbb{R}^{d \times r}$ and $\mathbf{A} \in \mathbb{R}^{r \times k}$. The final weight matrix $\mathbf{W}$ is computed as the sum of the initial weights $\mathbf{W}_0$ and the product of these low-rank matrices $\mathbf{BA}$.

The combined loss function integrates the TC loss with the standard task-specific loss:

$$\mathcal{L} = \mathcal{L}_{\text{Task}} + \lambda \mathcal{L}_{\text{TC}}, \qquad (12)$$

where $\lambda$ controls the influence of the TC regularization. During training, the LLM's hidden representations are affected by the updated weights, which are learned by minimizing this combined loss. This allows the LoRA updates to strategically direct the LLM toward achieving an optimally equidistant feature space configuration, thereby enhancing the discriminative power for underrepresented tail

classes while simultaneously maintaining parameter efficiency through minimal augmentation of trainable weights.

### 4.3. Theoretical Analysis

We conduct systematic gradient-based analysis of backpropagated loss signals to elucidate the mechanistic influence of our proposed methodology, specifically examining its optimization trajectory perturbations under imbalanced class distributions characteristic of real-world training scenarios.

**Gradient analysis**: The gradient of the task-specific loss $\mathcal{L}_{\text{Task}}$ with respect to text representations is:

$$\frac{\partial \mathcal{L}_{\text{Task}}}{\partial h_l(t_k)} = \sum_{i=1}^{n_k} (p_k(z_{k,i}) - 1) z_{k,i} + \sum_{k' \neq k} \sum_{j=1}^{n_{k'}} p_k(z_{k',j}) z_{k',j}. \quad (13)$$

This gradient consists of an "intra-class cohesion" term and an "inter-class repulsion" term. Under class-imbalanced learning paradigms, gradient signals for underrepresented minority classes become quantitatively dominated by the repulsion term, thereby creating geometric configuration biases that may result in suboptimal feature space organization for rare categories. The gradient of TC loss is:

$$\frac{\partial \mathcal{L}_{\text{TC}}}{\partial t_i} = \sum_{j=1, j \neq i}^{K} 2 \left( \langle h_l(t_i), h_l(t_j) \rangle - T_W \cdot \mu \right) \cdot \frac{\partial h_l(t_i)}{\partial t_i} h_l(t_j). \quad (14)$$

The TC loss explicitly adjusts text representations to ensure maximal angular separation, mitigating the adverse effects of imbalanced gradient updates. **Impact on class imbalance**: The TC loss provides an additional regularization term that is independent of class frequencies. This helps to balance the gradient updates across all classes:

(1) For minor classes, it compensates for the dominance of the inter-class repulsion term in $\mathcal{L}_{\text{Task}}$.

(2) For major classes, it ensures their representations do not overly dominate the embedding space.

By integrating the TC loss into the LoRA fine-tuning process, we encourage the model to learn a more equitable representation space. This theoretical analysis supports our method's effectiveness in addressing the long-tail distribution problem in educational data.

## 5. Experiments

In this section, we present a comprehensive evaluation of NCAL through extensive experiments on large language models using LoRA (Hu et al., 2021) parameter-efficient fine-tuning. Our investigation aims to address the following research questions: **Q1**: How does NCAL perform compared to current state-of-the-art methods in addressing long-tail data distributions (see Section 5.2)? **Q2**: How do different data distribution ratios affect the ETF structure and TC regularization in models? What is the relationship between data imbalance and the effectiveness of our proposed

| (a) TIMSS Math Word Problem Labeling (TMWPL) | (b) Productive Math Tutoring Dialogue (PMTD) |
|---|---|
| **Recall** 
 The sum of the interior angles of an obtuse triangle is the same as the sum of the interior angles of an acute triangle.   A. ×   B. √ 
 **Formulate** 
 In integer division, when two numbers are divided, the quotient is 31 and the remainder is 6. 
 What is the smallest divisor?  What is the dividend in this case? 
 **Identify** 
 If a month's income is represented by a positive number and ...... negative number, then Wang Xiaoqiang's income of 2,800 ¥ can be recorded as _____ ¥, and his expenditure ...... 
 **Represent** 
 In the new rural reconstruction, a village built a path x meters long. The construction team repaired 35 meters a day. After y days of repair, there were still ___ meters left. If x=200, y=5, there were still ___ meters left. 
 **Inferences** 
 To make 9□875≈100,000, there are (    ) numbers that can be filled in the □.   A. 6   B. 5   C. 4 
 **Analyze** 
 The school needs to buy uniforms for 25 football players. The unit prices of the uniforms are 78 ¥/set, 85 ¥/set and 104 ¥/set. How much is the minimum ......? What is the maximum ......? 
 **Implement** 
 ...... | **Initiation-Questioning (I-Q)** 
 Could you calculate 5 plus 3? Tell me your calculation process. 
 **Initiation-Hints (I-H)** 
 The question ...... We are given Tom's age, as well as the age difference between the father and Tom, and between the father and the grandfather. Can we use the age relationships ...... to determine the grandfather's age? 
 **Response-Refuse to Response (R-RR)** 
 I have no ideas. 
 **Response-Simplistic Response (R-SR)** 
 Yes, or okay. 
 **Response-Factual Response (R-FR)** 
 It is a well-established fact that the interior angles of any triangle sum to 180 degrees. 
 **Feedback-Feeding back on performance (F-F)** 
 You're right. 
 **Feedback-Instructing (F-I)** 
 In this problem, we know ...... Since the grandfather is 30 years older than the father, the grandfather's age would be 35 + 30, which equals 65 years old. 
 **Unrelatedness (U)** 
 Hi. Could you see my shared screen? |

Figure 3: Examples of TMWPL dataset with seven classes and PMTD dataset with eight classes.

method (see Section 5.3)? **Q3**: How does the NCAL model perform under diverse settings (see Section 5.3)?

### 5.1. Experimental Setup

**Datasets.** Experiments are demonstrated on two personalized learning datasets. The TMWPL dataset is designed to evaluate students' cognitive abilities, while the PMTD dataset focuses on classifying teacher-student dialogue behaviors. Examples from each dataset are shown in Figure 3. As illustrated in Figure 4, both datasets exhibit a characteristic long-tail distribution in sample counts across categories.

**TMWPL.** The TMWPL dataset is developed based on the TIMSS (Trends in International Mathematics and Science Study) assessment framework, aiming to comprehensively evaluate students' mathematical cognitive abilities. It covers seven core cognitive dimensions defined in the TIMSS Mathematics Cognitive Domains: Recall, Formulate, Identify, Represent, Implement, Inferences, and Analyze. The dataset consists of math word problems designed for students in grades 3 to 6, systematically annotated by professional education experts. Each problem was independently labeled by three to five annotators, with the final label determined by majority voting. Among the dimensions, Recall has the highest number of samples (5,045), while Analyze has the fewest (216). The test set includes 50 samples per cognitive dimension, with the remaining data used for training. Detailed definitions and representative examples of

each cognitive category are provided in Appendix A.

**PMTD.** The PMTD dataset captures one-on-one instructional interactions between teachers and students, with a particular focus on guided teaching strategies used to support students in solving challenging math problems. Drawing on the IRF framework (Sinclair & Coulthard, 2013) and scaffolding theory (Van de Pol et al., 2010), we collaborated with experts from the Education University to design an analytical framework comprising three primary categories and eight subcategories: Teacher Initiation (I-Q, I-H), Student Response (R-RR, R-SR, R-FR), Teacher Feedback (F-I, F-F), and Unrelated Content (U). Two trained annotators conducted the labeling process, achieving an inter-rater reliability of 86% (Cohen's Kappa), with any disagreements resolved through discussion. Among all categories, R-FR (feedback-related student responses) is the most frequent, accounting for 1,014 samples, while R-RR (repetition responses) is the least frequent, with only 62 samples. The test set includes 40 samples for each category, and the remaining data is used for training. Full definitions and illustrative examples are provided in Appendix B.

**Baselines.** In our evaluation of various large language models fine-tuned with LoRA to assess our approach, we have primarily selected three categories of models. This selection was guided by our dataset, which predominantly consists of Chinese content and focuses on the domain of mathematics. The models chosen for our study include: (1) Chinese-

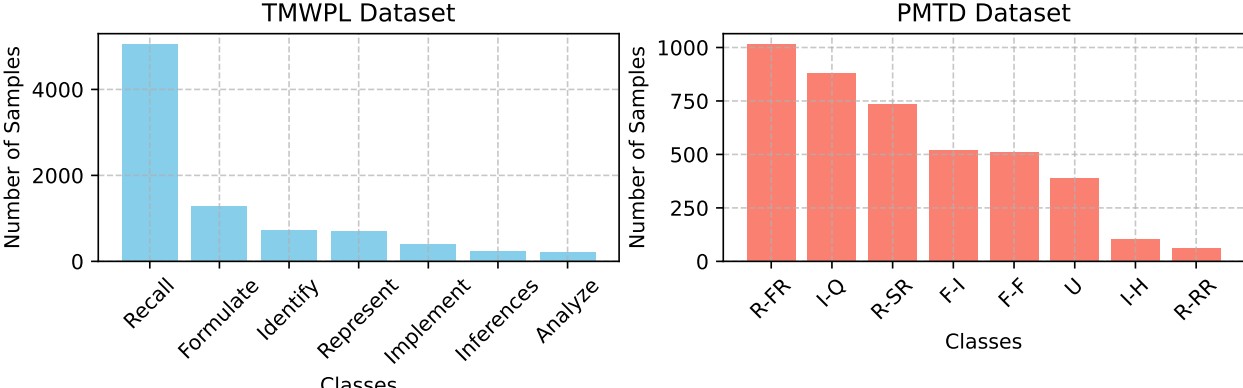

Figure 4: Sample distribution across categories in TMWPL and PMTD. A clear long-tailed distribution is observed in both.

focused instruction models: Qwen2.5 instruct (7B, 14B) and DeepSeek v2 chat. (2) English-focused instruction models: Llama3.1 instruct and Vicuna1.5 (7B, 13B). (3) Logic and math-oriented models: Qwen2.5-math and DeepSeek-math.

**Implementation details.** Our model training implements LoRA-based PEFT, using the LLaMA Factory (Zheng et al., 2024) automated training and inference pipeline as a foundation. All experiments were conducted on a system with eight NVIDIA A100 GPUs (80GB each).

## 5.2. Overall Performance

In Table 1, we report the overall performance of our NCAL and baselines on two long-tail datasets. According to Table 1, we have the following findings: (1) Our NC algorithm demonstrates consistently superior performance across Chinese and English large language models, including those specifically enhanced for reasoning tasks. (2) Our 7B NCAL model achieves remarkable results by outperforming models ranging from 7B to 14B, including both dense and Mixture-of-Experts (MoE) architectures. This notable achievement across all evaluated benchmarks demonstrates that our approach can deliver state-of-the-art performance while maintaining computational efficiency through a smaller parameter. (3) We also observed an interesting phenomenon: reasoning-specialized models excel in mathematical problem classification tasks, whereas instruction-tuned models outperform in dialogue classification scenarios. This finding underscores the importance of task-specific model optimization for achieving optimal performance.

## 5.3. Ablation Study

To validate the design of NCAL, we conducted ablation study focusing on data ratio configurations, the use of Base or Instruct-tuned LLMs, and the configuration of prompts.

**Data ratio analysis.** We examined data ratios ranging from 0.03 (minimum to maximum class quantity ratio) to 0.25,

with intermediate steps at 0.03, 0.15, and 0.25. Figure 5 illustrates that model accuracy improves with more balanced data distributions. However, we observed a performance decline at the 0.25 ratio due to insufficient data volume for effective fine-tuning. Our analysis reveals a strong correlation between the $\Delta TCD$ values and classification accuracy across all model variants. Specifically, larger $\Delta TCD$ values consistently correspond to lower accuracy scores, indicating that the degree of deviation from the ideal ETF structure directly impacts model performance. This finding suggests that maintaining a more uniform text representation structure is crucial for achieving optimal classification results, particularly in long-tail scenarios.

**Base or instruct-tuned LLMs.** We further investigated the performance differences between the base model and the instruction-tuned model. As shown in Table 2, the instruction-tuned models consistently outperformed the base model on both test datasets, achieving higher accuracy. Moreover, even after applying LoRA fine-tuning to our dataset, the base model still exhibited a pronounced tendency to generate uncontrolled and meaningless outputs. Therefore, we ultimately adopted an evaluation strategy that selects only the top N tokens. This experiment further validates our motivation: optimizing models on long-tail data may introduce negative feedback, highlighting the importance of properly handling long-tail samples during training.

**Configuration of prompts.** In the context of instruction-tuned models, the design of prompts significantly impacts their zero-shot capabilities, particularly in distinguishing between directly generating categories and generating categories after applying Chain of Thought (COT) reasoning. To address this issue, we investigated the influence of including a prompt that specifies classification tasks on model performance. Specifically, the optimal model input combined task descriptions with dataset features, while the scenario without prompts only included the dataset features (detailed prompt configurations can be found in Appendix C). Our results in Tabel 2 indicated that, in the scenario where only

Table 1: Experiments on the two long-tail datasets, TMWPL and PMTD. The best results are highlighted in bold.

| | | | TMWPL | | PMTD | |
|---|---|---|---|---|---|---|
| Model | Type | Parameters | Acc | Macro-F1 | Acc | Macro-F1 |
| Qwen2.5-Instruct (Qwen et al., 2025) | Chinese-focused | 7B | 61.14 | 60.59 | 65.94 | 61.46 |
| Qwen2.5-Instruct (Qwen et al., 2025) | Chinese-focused | 14B | 65.14 | 65.02 | 66.56 | 61.91 |
| DeepSeek-V2-Lite-Chat (Liu et al., 2024) | Chinese-focused | 16B | 65.14 | 66.17 | 61.56 | 53.78 |
| NCAL-Qwen2.5-Instruct | Chinese-focused | 7B | **74.86** | **75.58** | **70.31** | **67.43** |
| Vicuna1.5 (Chiang et al., 2023) | English-focused | 7B | 63.14 | 62.61 | 61.88 | 53.55 |
| Llama3.1-Instruct (Dubey et al., 2024) | English-focused | 8B | 61.43 | 61.73 | 65.62 | 60.41 |
| Vicuna1.5 (Chiang et al., 2023) | English-focused | 13B | 67.43 | 68.26 | 62.50 | 56.31 |
| NCAL-Llama3.1-Instruct | English-focused | 8B | **69.43** | **69.14** | **67.50** | **64.92** |
| Qwen2.5-Math-Instruct (Qwen et al., 2025) | Logic and math-oriented | 7B | 67.71 | 68.77 | 59.69 | 51.14 |
| Deepseek-Math-Instruct (Liu et al., 2024) | Logic and math-oriented | 7B | 67.71 | 68.20 | 59.38 | 51.37 |
| NCAL-Qwen2.5-Math-Instruct | Logic and math-oriented | 7B | **74.86** | **75.51** | **63.12** | **58.32** |

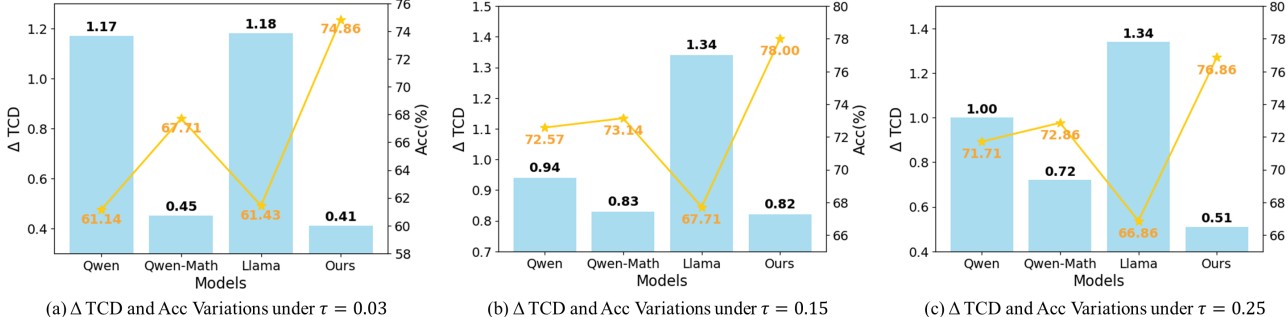

(a) ∆ TCD and Acc Variations under $\tau = 0.03$    (b) ∆ TCD and Acc Variations under $\tau = 0.15$    (c) ∆ TCD and Acc Variations under $\tau = 0.25$

Figure 5: Comparison of $\Delta$TCD with model performance on the TMWPL dataset, where $\Delta$TCD is calculated based on $\mathcal{L}_{\text{TC}}$. $\tau$ indicates the degree of data imbalance, with $\tau = 0.25$ for less imbalance and $\tau = 0.03$ for high imbalance. Our findings show that, overall, as data balance improves, classification performance is significantly enhanced.

Table 2: Ablation study of prompt in NCAL and LLMs.

| Model | TMWPL-Acc | PMTD-Acc |
|---|---|---|
| Qwen2.5-7B-Instruct | 74.86 | 70.31 |
| w/o instruction sft | 74.00 (0.86 ↓) | 70.00 (0.31 ↓) |
| w/o prompt | 75.00 (0.14 ↑) | 67.19 (3.12 ↓) |
| w/o prompt & NCAL | 73.71 (1.15 ↓) | 62.50 (7.81 ↓) |
| Llama3.1-8B-Instruct | 69.43 | 67.50 |
| w/o instruction sft | 68.29 (1.14 ↓) | 66.87 (0.63 ↓) |
| w/o prompt | 69.71 (0.28 ↑) | 62.19 (5.31 ↓) |
| w/o prompt & NCAL | 67.14 (2.29 ↓) | 56.25 (11.25 ↓) |
| Qwen2.5-Math-7B-Instruct | 74.86 | 63.12 |
| w/o instruction sft | 72.00 (2.86 ↓) | 62.50 (0.62 ↓) |
| w/o prompt | 71.43 (3.34 ↓) | 60.31 (2.81 ↓) |
| w/o prompt & NCAL | 71.43 (3.34 ↓) | 58.44 (4.68 ↓) |

the dataset feature inputs without any extra prompts underwent LoRA fine-tuning, all models exhibited a notable decline in performance. However, after applying our proposed NCAL method, instruction-related models showed a slight improvement in the TMWPL dataset. Despite a minor decline in performance on the PMTD dataset, our NCAL method generally demonstrated relative stability under the influence of varying prompts.

## 6. Conclusion

In this paper, to the best of our knowledge, we are the first paper to study the long-tail problem for data-based personalized learning. To address the real-word challenge, we propose **neural-collapse-advanced personalized Learning (NCAL)**. NCAL optimizes feature representations to conform to the simplex ETF structure and introduces Text-modality Collapse (TC) regularization to enhance the distribution of text embeddings within the LLM representation space. Through comprehensive experiments on two benchmark datasets (TMWPL and PMTD), we demonstrate three key findings: (1) NCAL exhibits consistent superior performance across various LLMs, including both Chinese and English models, validating its versatility; (2) Our 7B parameter NCAL achieves state-of-the-art performance while maintaining computational efficiency, outperforming larger models up to 14B parameters; (3) The effectiveness of NCAL remains stable across different prompt configurations and model architectures. Additionally, our ablation study confirms that NCAL effectively mitigates class imbalance and enhances model generalization ability, making it a promising solution for practical personalized learning.

## Acknowledgements

This work was supported in part by the National Natural Science Foundation of China (No. 62477012) and the Natural Science Foundation of Shanghai (No. 23ZR1418500). The authors would like to express sincere gratitude to these funding agencies for their support. The opinions and conclusions presented in this paper are solely those of the authors and do not necessarily reflect the views of the funding organizations. Sen Cui would like to acknowledge the financial support received from the Shuimu Tsinghua scholar program.

## Impact Statement

This paper presents work whose goal is to advance the field of Machine Learning. There are many potential societal consequences of our work, none which we feel must be specifically highlighted here.

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

# Appendix

We provide supplementary details to enhance the understanding of our research. Specifically, it focuses on dataset class descriptions and experimental prompt design, offering concise insights into the methodological nuances and experimental setup that support our main paper's findings.

## A. The TMWPL Dataset label description

Table 3: Mathematics Cognitive Domains

| Level | Label Type | Label Description |
|---|---|---|
| A | Recall | Recall definitions, terminology, number properties, units of measurement, geometric properties, and notation (e.g., $a \times b = ab$, $a + a + a = 3a$). |
| B | Formulate | Determine efficient/appropriate operations, strategies, and tools for solving problems. |
| C | Identify | Identify numbers, expressions, quantities, and shapes Recognize when entities are mathematically equivalent Read information from graphs, tables, texts, or other sources. |
| D | Represent | Represent data in tables or graphs; create equations, inequalities, geometric figures, or diagrams that model problem situations; and generate equivalent representations for a given mathematical entity or relationship. |
| E | Implement | Implement suitable strategies and operations to produce solutions to problems. |
| F | Inferences | Make valid inferences based on information and evidence. |
| G | Analyze | Analyze, describe, or use relationships among numbers, expressions, quantities, and shapes. |

## B. The PMTD Dataset label description

Table 4: Dialogue Actions, Definitions, and Examples

| Dialogue Level | Dialogue Action | Definition | Example |
|---|---|---|---|
| Initiation (I) | Questioning (I-Q) | It involves asking students questions that require an active linguistic and cognitive answer. | Could you calculate 5 plus 3? Tell me your calculation process. |
| | Hints (I-H) | It entails the provision of clues or suggestions by the teacher to help the student go forward. The teacher deliberately does not supply the entire solution or detailed instructions under such circumstances. | The question asks us to calculate the grandfather's age. We are given Tom's age, as well as the age difference between the father and Tom, and between the father and the grandfather. Can we use the age relationships between the father and Tom, and the father and the grandfather, to determine the grandfather's age? |
| Response (R) | Refuse to Response (R-RR) | Refuse to answer questions or remain silent without providing a response. | I have no ideas or choose to remain silent. |
| | Simplistic Response (R-SR) | The responses are simple to the point of lacking depth or thoroughness. | Mm, yes, or okay. |
| | Factual Response (R-FR) | The answers provided are characterized by their factuality, reliance on memory, and explanatory nature. | It is a well-established fact that the interior angles of any triangle sum to 180 degrees. |
| Feedback (F) | Feeding back on performance (F-F) | It involves the provision of information regarding the student's performance to the student him/herself. | You're right / You're smart. |
| | Instructing (F-I) | It involves the teacher telling the students what to do or explanation of how something must be done and why. | In this problem, we know the age difference between the father and Tom is 30 years. By taking Xiao Tom's age, which is 5 years old, and adding 30 years, we find that the father is currently 35 years old. Since the grandfather is 30 years older than the father, the grandfather's age would be 35 + 30, which equals 65 years old. |
| Unrelatedness (U) | Unrelatedness (U) | It does not directly relate to teaching and learning OR the topic under study. | Hi. Could you see my shared screen? |

## C. Prompt settings

---

**Prompts of TMWPL Tuning.**

Now there is a problem with the content [ **question** ], please determine which category this problem belongs to from the following categories:

- **Recall**: Recall definitions, terminology, number properties, units of measurement, geometric properties, and notation (e.g., a × b = ab, a + a + a = 3a).
- **Formulate**: Determine efficient/appropriate operations, strategies, and tools for solving problems.
- **Identify**: Identify numbers, expressions, quantities, and shapes Recognize when entities are mathematically equivalent Read information from graphs, tables, texts, or other sources.
- **Represent**: Represent data in tables or graphs; create equations, inequalities, geometric figures, or diagrams that model problem situations; and generate equivalent representations for a given mathematical entity or relationship.
- **Implement**: Implement suitable strategies and operations to produce solutions to problems.
- **Inferences**: Make valid inferences based on information and evidence.
- **Analyze**: Analyze, describe, or use relationships among numbers, expressions, quantities, and shapes.

**Note** Please directly output the category of the target problem, with no additional output.

---

Figure 6: Example Prompts of Model Tuning in TMWPL Dataset.

---

**Prompts of PMTD Tuning.**

There is a dialogue between a teacher and a student (a small part extracted from it). Please classify the target dialogue according to the classification requirements.

### Partial Dialogue Context

**Student:** XXXX.
**Tutoring Teacher:** XXXX?
**Student:** XXX!

### Dialogue to be Classified

**Tutoring Teacher:** XXX?

### Classification Requirements

- **I-Q**: It involves asking students questions that require an active linguistic and cognitive answer.
- **I-H**: It entails the provision of clues or suggestions by the teacher to help the student go forward. The teacher deliberately does not supply the entire solution or detailed instructions under such circumstances.
- **F-F**: It involves the provision of information regarding the student's performance to the student him/herself.
- **F-I**: It involves the teacher telling the students what to do or explanation of how something must be done and why.
- **R-RR**: Refuse to answer questions or remain silent without providing a response.
- **R-SR**: The responses are simple to the point of lacking depth or thoroughness.
- **R-FR**: The answers provided are characterized by their factuality, reliance on memory, and explanatory nature.
- **U**: It does not directly relate to teaching and learning OR the topic under study.

### Note

Please directly output the type of the target dialogue, with no additional output.

---

Figure 7: Example Prompts of Model Tuning in PMTD Dataset.

