# OpenReview forum: "Advancing Personalized Learning with Neural Collapse for Long-Tail Challenge"
_ICML.cc/2025/Conference — ICML 2025 poster_

### Official Review · Reviewer_zV7k · 2025-03-08

**Overall Recommendation:** 4

**Summary:**

This paper addresses a major challenge in personalized learning, where many existing methods assume high-quality, well-annotated benchmarks. In real-world settings, such benchmarks often exhibit long-tail distributions that negatively affect model performance. The authors propose a novel approach called Neural-Collapse-Advanced Personalized Learning (NCAL), which introduces a Textmodality Collapse (TC) regularization to optimize the distribution of text embeddings in large language models (LLMs). NCAL is model-agnostic, ensuring it can work with various architectures, and it is shown to improve performance significantly, surpassing previous state-of-the-art methods while mitigating class imbalance.

**Claims And Evidence:**

The paper demonstrates convincing evidence that NCAL effectively addresses the long-tail distribution issue and enhances performance. The claim about improving generalization ability is well-supported by experiments. However, the text lacks detailed information on the underlying mathematical principles of the Textmodality Collapse regularization, and a deeper explanation would be beneficial.

**Essential References Not Discussed:**

In my understanding, the important references have already been discussed.

**Experimental Designs Or Analyses:**

The authors conducted experiments on two real-world personalized learning datasets and compared various types of large language models as baselines. The overall experimental design is reasonable and sufficiently demonstrates the effectiveness of the proposed method. However, the authors should visualize the statistical analysis of these two real-world datasets.

**Methods And Evaluation Criteria:**

Yes, the proposed method and evaluation criteria make sense for the problem. Personalized learning is a significant area of interest, and addressing the limitations of the benchmark, especially their long-tail distributions. It is an important step in improving model performance. The introduction of NCAL, with its Textmodality Collapse regularization, is a reasonable approach to optimize text embeddings and improve generalization ability.

**Other Comments Or Suggestions:**

In Equation 5, the symbol $\alpha$ is not explained.

**Other Strengths And Weaknesses:**

## Strengths

- NCAL enhances the generalization ability of models by addressing the challenge of class imbalance, improving performance on diverse datasets.
- By being model-agnostic, NCAL can be applied across different architectures, increasing its versatility and potential for wide adoption in various domains.

## Weaknesses

- The authors should provide a statistical analysis of the datasets used in the experiments and visually present the long-tail distribution of sample sizes in each category for a clearer understanding.

- In Figure 4, the authors show the performance differences between various methods based on different values of $\tau$. The rationale for selecting $\tau = 0.25/0.03/0.15/0.25$ is unclear. The authors should explain why these four specific values of $\tau$ were chosen for the experiments.

- In Table 3, two performance spikes are observed for the “w/o prompt” condition in green. The authors should provide an explanation for the possible reasons behind this unexpected result.

**Questions For Authors:**

See other comments.

**Relation To Broader Scientific Literature:**

This paper presents NCAL, a model-agnostic method that addresses long-tail distributions in personalized learning by optimizing text embeddings with Textmodality Collapse (TC) regularization. NCAL improves model generalization, mitigates class imbalance, and achieves state-of-the-art performance, advancing data-driven methods in real-world scenarios.

**Theoretical Claims:**

I have specifically checked the correctness of the proofs for the theoretical claims in this paper. However, the TC loss in Equation 14 and the loss in Equation 13 should be further explained and clarified.

---

> ### Author Rebuttal · Authors · 2025-04-01
>
> > **Q1: However, the text lacks detailed information on the underlying mathematical principles of the Textmodality Collapse regularization, and a deeper explanation would be beneficial.**
>
> **A1:** Thank you for the reviewer's insightful comment. In our paper, the goal of TC regularization is to promote a more balanced and structured text feature distribution, thereby addressing the long-tail distribution problem by leveraging the constraints from the information of different categories. In Section 4.3, we explain the mathematical principles behind TC regularization from the perspective of gradient optimization and its role in constraining the text feature space.
>
> > **Q2: However, the TC loss in Equation 14 and the loss in Equation 13 should be further explained and clarified.**
>
> **A2:** We further clarify the distinction between the TC loss in Equation 14 and the loss in 13. The gradient in 13 consists of an "intra-class cohesion" term and an "inter-class repulsion" term. In class-imbalanced learning paradigms, the gradient signals for underrepresented minority classes become dominated by the repulsion term, which creates geometric configuration biases that can result in suboptimal feature space organization for rare categories. Equation (14) explicitly shows that the TC loss adjusts text representations to ensure maximal angular separation, mitigating the adverse effects of imbalanced gradient updates.
>
> > **Q3: However, the authors should visualize the statistical analysis of these two real-world datasets.**
>
> **A3:** In the TMWPL dataset, there are seven classes: Recall, Formulate, Identify, Represent, Implement, Inferences, and Analyze, with sample sizes of 5045, 1283, 717, 711, 405, 235, and 216, respectively. On the other hand, the PMTD dataset consists of eight classes: R-FR, I-Q, R-SR, F-I, F-F, U, I-H, and R-RR, with sample sizes of 1014, 880, 737, 518, 510, 389, 104, and 62, respectively.
>
> > **Q4: The authors should explain why these four specific values of $\tau$ were chosen for the experiments.**
>
> **A4:** Firstly, $ \tau $ represents the imbalance ratio of the data, defined as the ratio of the number of samples in the least frequent category to the number of samples in the most frequent category. In our paper, 0.03 corresponds to the $ \tau $ of our original data, while 0.25 is the maximum $ \tau $ that our data can support for training without compromising the model's performance. Beyond this threshold, the data volume is insufficient for effective training.
>
> > **Q5:** The authors should explain the possible reasons behind this unexpected result.
>
> **A5:** The reviewer raised an interesting question. In the ablation experiments, the performance improvement of the "w/o prompt" condition compared to the baseline method may stem from the model's ability to more effectively construct the text feature space in the absence of a prompt. This suggests that removing the prompt could allow the model to be more flexible in feature extraction, thereby enhancing performance. Although this phenomenon is intriguing, we believe further experiments are needed to validate its correctness, and we plan to explore this as part of future work.
>
> > **Q6: More experimental designs for our method.**
>
> **A6:** We followed the reviewer's suggestion and considered additional baselines to compare the effectiveness of our method. The experimental results are shown below. Under the comparison with the same model parameters and even larger model parameters, our method still demonstrates robust performance on long-tail distribution datasets. This further confirms that the introduction of neural collapse can constrain the learning of the model's class space, thus avoiding biases caused by data with a large number of samples.
>
> | Model | Parameters | Dataset | Acc |
> | --- | --- | --- | --- |
> | glm-4-9b-chat  | 9B | TMWPL | 67.71 |
> | Ministral-8B-Instruct | 8B | TMWPL | 66.86 |
> | Yi-1.5-9B-Chat | 9B | TMWPL | 63.14 |
> | Baichuan2-7B-Chat | 7B | TMWPL | 61.14 |
> | **NCAL-Qwen2.5-Instruct**   | 7B  | TMWPL | **74.86** |
>
> > **Q7: In Equation (5), the symbol $\alpha$ is not explained.**
>
> **A7:** $\alpha$ is a constant parameter for scaling in our paper.

---

> > ### Comment · Reviewer_zV7k · 2025-04-05
> >
> > Thanks to the authors for the rebuttal. After carefully reviewing the authors' responses, all of my concerns have been adequately addressed, including a detailed rationale for the proposed technique and a more in-depth experimental analysis. Therefore, I am willing to raise my rating.

---

> > > ### Author Response · Authors · 2025-04-05
> > >
> > > We are glad to know that your concerns have been effectively addressed. We are very grateful for your constructive comments and questions, which helped improve the clarity and quality of our paper. Thanks again!

---

### Official Review · Reviewer_jYxM · 2025-03-10

**Overall Recommendation:** 3

**Summary:**

Personalized learning, particularly data-driven approaches, faces challenges due to long-tail distributions in real-world benchmarks, which affect model performance. To address this, the authors propose NCAL (Neural-Collapse-Advanced Personalized Learning), which leverages Textmodality Collapse (TC) regularization to optimize text embeddings in large language models (LLMs). NCAL is model-agnostic, compatible with various architectures, and improves generalization while mitigating class imbalance. Extensive experiments show that NCAL achieves state-of-the-art performance and enhances existing methods.

**Claims And Evidence:**

The claims in the submission are generally well-supported by clear evidence, with experiments demonstrating NCAL’s effectiveness in addressing long-tail distributions and class imbalance. However, the claim about mitigating class imbalance would benefit from further discussion or visualizations to highlight the improvement in class distribution. More visual evidence could strengthen the claim of enhanced category representation.

**Essential References Not Discussed:**

The paper provides a solid discussion of related works.

**Experimental Designs Or Analyses:**

The experimental design in this paper is generally sound, with a clear setup and methodology. However, incorporating additional experiments to evaluate the method’s performance under varying imbalance rates and providing more details on the experimental setup, such as dataset characteristics and imbalance ratios, would offer deeper insights into the model’s robustness and generalizability.

**Methods And Evaluation Criteria:**

Yes, the proposed methods and evaluation criteria are appropriate for the problem. The use of NCAL to address the long-tail distribution problem in personalized learning is a novel approach. The experiments on benchmark datasets (TIMSS and PMTD) are suitable for evaluating the effectiveness of the proposed method in real-world scenarios.

**Other Comments Or Suggestions:**

Please refer to other reviews.

**Other Strengths And Weaknesses:**

## Strengths
- The paper introduces NCAL, which leverages the simplex ETF structure and TC regularization to mitigate class imbalance, improving model generalization in personalized learning.

- The proposed method is model-agnostic and demonstrates superior performance across multiple architectures and benchmark datasets, achieving state-of-the-art results.


## Weaknesses
- Could you further explain Equation (10)? Are $i$ and $j$ texts from the same category or different categories?

-  Although the authors have validated the effectiveness of their method on two representative data-driven personalized learning datasets, it is recommended to conduct experiments on additional open-source datasets to more comprehensively assess the method’s generalization ability and robustness.

- Table 3 shows the impact of the presence or absence of prompts on performance. Have the authors considered the specific impact of different types of prompts on performance?

**Questions For Authors:**

Please refer to the weaknesses.

**Relation To Broader Scientific Literature:**

This paper builds upon prior research in personalized learning and long-tail distribution challenges, addressing limitations in existing benchmarks that assume high-quality, well-annotated data. By leveraging the simplex ETF structure and introducing Textmodality Collapse (TC) regularization, NCAL refines text embeddings within LLMs, aligning with recent advancements in feature representation learning.

**Theoretical Claims:**

Yes, I have reviewed the theoretical claims in the paper. The proofs for the claims related to the effectiveness of the NCAL method appear to be logically sound and well-supported by the experiments. However, the mathematical formulation and the justification behind the introduction of Textmodality Collapse (TC) regularization could benefit from further explanation, particularly regarding its impact on the feature representations and how it mitigates the long-tail distribution issue. More clarification in this regard would strengthen the theoretical foundation of the approach.

---

> ### Author Rebuttal · Authors · 2025-04-01
>
> > **Q1: However, the claim about mitigating class imbalance would benefit from further discussion or visualizations to highlight the improvement in class distribution. More visual evidence could strengthen the claim of enhanced category representation.**
>
> **A1:** We appreciate the reviewer’s comments. Below, we present the statistical distribution of the data used in our paper. Please note that these two datasets were collected from real-world scenarios, with PMTD containing eight categories and TMWPL comprising seven categories.
>
> | Classes number  | Class 1 | Class 2  | Class 3 | Class 4 | Class 5  | Class 6 | Class 7  | Class 8 |
> | ------- | ---- | ---- | ---- | ---- | ---- | ---- | ---- | ---- |
> | Classes  | R-FR | I-Q  | R-SR | F-I  | F-F  | U    | I-H  | R-RR |
> | PMTD samples | 1014 | 880  | 737  | 518  | 510  | 389  | 104  | 62   |
> | Classes  | Recall | Formulate | Identify | Represent | Implement | Inferences | Analyze |
> | TMWPL samples | 5045   | 1283    | 717   | 711  | 405   | 235  | 216 |
>
> > **Q2: However, the mathematical formulation and the justification behind the introduction of TC regularization could benefit from further explanation, and how it mitigates the long-tail distribution issue.**
>
> **A2:** The goal of NCAL is to learn a simple ETF through the method of neural collapse, to construct a uniformly partitioned category feature space. To achieve this, we introduce TC regularization, which aims to constrain the feature distribution of category partitioning learned from text data across different categories, effectively addressing the long-tail distribution problem.
>
> > **Q3: However, incorporating additional experiments to evaluate the method’s performance under imbalance rates and experimental details.**
>
> **A3:** On the TMWPL dataset, we conducted experiments based on the Qwen2.5-Math-Instruct architecture, evaluating the model's performance under different levels of long-tail distribution, ranging from low to high imbalance ratios (0.03, 0.08, 0.13, 0.18, 0.25). The corresponding performance results were 74.86, 75.01, 76.45, 78.06, and 76.86. The experimental results indicate that as the degree of long-tail distribution weakens, the model's performance improves.
>
> > **Q4: Could you further explain Equation (10)? Are i and j texts from the same category or different categories?**
>
> **A4:** Yes, I will provide a more detailed explanation for Equation (10). The texts $i$ and $j$ come from different categories, and the goal is to constrain the feature representations of samples from different categories to approach a uniform distribution, i.e., $1/(N-1)$, where $N$ represents the number of categories.
>
> > **Q5: Although the authors have validated the effectiveness of their method on two representative data-driven personalized learning datasets, it is recommended to conduct experiments on additional open-source datasets to more comprehensively assess the method’s generalization ability and robustness.**
>
> **A5:** As recommended by the reviewer, we have included a comprehensive study comparing three open-source datasets: [IITJEE NEET AIIMS Students Questions Data](https://www.kaggle.com/datasets/mrutyunjaybiswal/iitjee-neet-aims-students-questions-data), [MathDial](https://github.com/eth-nlped/mathdial/tree/main), and [DialogID](https://github.com/ai4ed/DialogID/tree/main). The results, presented below, highlight the effectiveness of NCAL compared to additional baselines. The experimental results indicate that our method consistently outperforms others across all datasets.
>
> | Model | Qwen2.5-Instruct | NCAL-Qwen2.5-Instruct | Llama3.1-Instruct | NCAL-Llama3.1-Instruct | Qwen2.5-Math-Instruct | NCAL-Qwen2.5-Math-Instruct |
> |-------|-------|-------|-------|-------|-------|-------|
> | Parameters | 7B | 7B | 8B | 8B | 7B | 7B |
> | INASQD-Acc | 94.50 | **96.50** | 95.50 | **98.00** | 95.00 | **96.50** |
> | MathDial-Acc | 51.50 | **57.00** | 51.00 | **53.50** | 52.00 | **57.50** |
> | DialogID-Acc | 85.15 | **87.46** | 85.75 | **87.34** | 84.53 | **86.03** |
>
> > **Q6:** The impact of different types of prompts on performance?
>
> **A6:** Thank you for the reviewer’s suggestion. Based on the prompts considered in our manuscript, we have referred to related work and explored additional prompt forms, including base prompt (see Table 5 in our manuscript), w/o prompt, few-shot (introducing a few reference samples in the prompt), role play (personalized scene prompts), and only class name. The experimental results are shown below. The results demonstrate that our method maintains strong robustness and generalization across different prompt settings.
>
> | Model | base prompt | w/o prompt | few-shot | role play | only classname |
> | -- | -- | -- | -- | -- | -- |
> | NCAL-Qwen2.5-Instruct | 74.86 | 75.00 | 74.86 | 74.86 | 74.00 |
> | NCAL-Llama3.1-Instruct     | 69.43 | 69.71 | 68.29 | 68.57 | 69.43 |
> | NCAL-Qwen2.5-Math-Instruct | 74.86 | 71.43 | 72.57 | 74.86 | 72.00 |

---

### Official Review · Reviewer_kS9o · 2025-03-10

**Overall Recommendation:** 4

**Summary:**

Personalized learning has gained significant attention due to its ability to address individual student needs. Still, many methods rely on the assumption of high-quality benchmarks, which are often unrealistic. To overcome this, the authors proposed NCAL, which utilizes a TC regularization to adjust the distribution of text embeddings within LLMs. NCAL is designed to be compatible with multiple models and shows impressive results in improving generalization and overcoming class imbalance. Experiments confirmed its ability to achieve SOTA performance.

**Claims And Evidence:**

While the paper provides evidence for the performance improvements and generalization ability of NCAL, the claim of it being “model-agnostic” remains somewhat unclear. More specific details on its application to different models and a deeper explanation of how it addresses class imbalance will help to reinforce these claims.

**Essential References Not Discussed:**

no

**Experimental Designs Or Analyses:**

yes

**Methods And Evaluation Criteria:**

yes

**Other Comments Or Suggestions:**

Please refer to Other Strengths And Weaknesses

**Other Strengths And Weaknesses:**

## Strengths
- The proposed method can effectively tackle the problem of long-tail distributions in real-world data, which often affects the performance of models, making the method highly relevant for practical applications.

- NCAL shows strong results in improving existing models, achieving state-of-the-art performance while also enhancing the model’s ability to deal with class imbalance.


## Weaknesses
- The authors should add ablation experiments to further discuss the impact of different imbalance ratios.

- Different forms of prompts can significantly impact the performance of LLM models. In Section 5.3, the authors examined the effect of using or not using prompts on the experimental results but did not further analyze the performance differences across different prompt formats.

- I believe the authors are tackling an important and practical issue in the field of personalized learning. I recommend including more baseline methods，e.g., [1-3]， for data-driven personalized learning to further strengthen the comprehensiveness of the study.

[1] Askarbekuly, N. and Aniciˇ c, N. Llm examiner: automating assessment in informal self-directed e-learning using chatgpt. Knowledge and Information Systems, pp. 1–18,2024.

[2] Ayeni, O. O., Al Hamad, N. M., Chisom, O. N., Osawaru,B., and Adewusi, O. E. Ai in education: A review of personalized learning and educational technology. GSC Advanced Research and Reviews, 18(2):261–271, 2024.

[3] Chang, L.-H. and Ginter, F. Automatic short answer grading for finnish with chatgpt. In Proceedings of the AAAI Conference on Artificial Intelligence, volume 38, pp. 23173–23181, 2024.

**Questions For Authors:**

In Section 6, the concept of equidistant text representation (ETR) regularization needs further explanation. Does it share the same meaning as Textmodality Collapse (TC)?

**Relation To Broader Scientific Literature:**

The method contributes to the broader scientific literature by addressing the long-tail problem in data-based personalized learning.

**Theoretical Claims:**

yes, no issues found

---

> ### Author Rebuttal · Authors · 2025-04-01
>
> > **Q1: While the paper provides evidence for the performance improvements and generalization ability of NCAL, the claim of it being “model-agnostic” remains somewhat unclear.**
>
> **A1:** NCAL enhances feature learning by aligning with the simplex equiangular tight frame (ETF) structure through Text-modality Collapse (TC) regularization, which optimizes text distribution. This design ensures NCAL's compatibility with various models, making it a versatile plug-and-play framework. As evidenced in Table 2, we demonstrate its effectiveness across multiple LLM architectures, including Qwen2.5 and Llama3.1, further supporting its model-agnostic property.
>
> > **Q2: More specific details on its application to different models and a deeper explanation of how it addresses class imbalance will help to reinforce these claims.**
>
> **A2:** We have further clarified the learning mechanism of NCAL. Specifically, NCAL aims to learn features that conform to the Equiangular Tight Frame (ETF) structure by introducing Text-Modality Collapse (TC) regularization. This optimization improves the distribution of text embeddings within the representation space of large language models (LLMs), thereby enhancing the robustness and generalization ability of LoRA fine-tuned LLMs under long-tail data distributions.
>
> > **Q3: The authors should add ablation experiments to further discuss the impact of different imbalance ratios.**
>
> **A3:** As suggested by the reviewer, we conducted a comprehensive comparative study on different imbalance ratios. Specifically, we evaluated the model under imbalance ratios of 0.03, 0.05, 0.10, 0.20, and 0.25. The experimental results are shown below, indicate that as the imbalance ratio decreases from 0.03 to 0.25, the model's performance improves accordingly, increasing from 74.86 to 76.86.
>
> | Data Redio (NCAL-Qwen2.5-Math-Instruct) | TMWPL-Acc |
> | - | - |
> | 0.03 | 74.86 |
> | 0.05 | 75.29 |
> | 0.10 | 77.14 |
> | 0.15 | 78.00 |
> | 0.20 | 78.29 |
> | 0.25 | 76.86 |
>
> > **Q4: Different forms of prompts can significantly impact the performance of LLM models. In Section 5.3, the authors examined the effect of using or not using prompts on the experimental results but did not further analyze the performance differences across different prompt formats.**
>
> **A4:** Following the reviewer's suggestion and related work, we validate our method with more prompt forms, including base prompt (see Table 5 in our manuscript), w/o prompt, few-shot (introducing a few reference samples in the prompt), role play (personalized scene prompts), and only class name（see Table 4）. The experimental results are shown below:
>
> | Model  | base prompt | w/o prompt | few-shot | role play | only class name |
> | ------- | ----------- | ---------- | -------- | ---------------- | -------------------- |
> | NCAL-Qwen2.5-Instruct      | 74.86       | 75.00      | 74.86    | 74.86 | 74.00  |
> | NCAL-Llama3.1-Instruct     | 69.43       | 69.71      | 68.29    | 68.57  | 69.43  |
> | NCAL-Qwen2.5-Math-Instruct | 74.86       | 71.43      | 72.57    | 74.86            | 72.00                |
>
> The experimental results show that different prompt forms have little impact on the model. The possible reason is that our method, using the TC regularization approach, can learn a good category feature space that is not overly influenced by the prompt during training and inference. This experiment further supports our motivation.
>
> > **Q5: I recommend including more baseline methods, for data-driven personalized learning to further strengthen the comprehensiveness of the study.**
>
> **A5:** We thank the reviewer for pointing out this issue. We conducted extensive experiments based on the papers recommended by the reviewer. Unfortunately, we found that the codes for the methods in [1] and [3] have not been released, making it difficult for us to reproduce these methods within the short time frame for the rebuttal. Additionally, [2] is a survey paper, from which we selected the most recent representative state-of-the-art (SOTA) works for performance comparison. The experimental results are as follows, showing that our method still achieves SOTA performance compared to other SOTA baselines under the more challenging TMWPL dataset.
>
> | Model | Llama3.1-Instruct|gemma-2-9b-it | internlm3-8b-instruct | **NCAL-Qwen2.5-Instruct**|
> | ------- | ---------- | --------- | --------- | --------|
> | Parameters | 8B | 9B | 8B | 7B |
> | Dataset | TMWPL | TMWPL | TMWPL| TMWPL|
> | Acc |61.43 |72.29|72.86| **74.86**|
>
>
> > **Q6: The concept of equidistant text representation (ETR) regularization needs further explanation. Does it share the same meaning as Textmodality Collapse (TC)?**
>
> **A6:** Thank you for the reviewer's comment. In our manuscript, the meaning of ETR is the same as TC, and we will correct this error in the final version.

---

> > ### Comment · Reviewer_kS9o · 2025-04-05
> >
> > The rebuttal have addressed most of my questions and concerns regarding the method and experiments. I have raised my score to 4 and recommend that the authors incorporate the relevant experiments and discussions on model generalization into the paper.

---

> > > ### Author Response · Authors · 2025-04-05
> > >
> > > We’re pleased to hear that your concerns have been resolved. We sincerely appreciate your thoughtful feedback and insightful suggestions, which have greatly contributed to enhancing the clarity and overall quality of our work. Thank you once again!

---

### Official Review · Reviewer_FPDE · 2025-03-10

**Overall Recommendation:** 4

**Summary:**

The paper proposes a new method called Neural-Collapse-Advanced Personalized Learning (NCAL) to address the limitations of data-based personalized learning approaches that assume well-annotated benchmarks. In reality, these benchmarks often have long-tail distributions that affect model accuracy. NCAL introduces a regularization technique, Textmodality Collapse (TC), to optimize text embedding distributions in LLMs. The method is model-agnostic, allowing for its integration with various architectures. The authors demonstrate that NCAL improves performance across tasks and mitigates class imbalance, achieving new state-of-the-art results in the field.

**Claims And Evidence:**

The claims regarding NCAL’s enhancement of model performance and its potential to address class imbalance are well-supported by experiments. However, the assertion that NCAL is model-agnostic would benefit from further clarification, particularly in terms of its practical application to various architectures. Additionally, more detailed examples of its impact on real-world class imbalance problems would provide further credibility to the claims.

**Essential References Not Discussed:**

As far as I understand, the key references have already been covered.

**Experimental Designs Or Analyses:**

The experimental design of this paper aligns with the standards of data-driven personalized learning in terms of data selection, baseline setup, and implementation details. However, incorporating additional personalized learning baselines or datasets would further enhance the completeness and comprehensiveness of experiments.

**Methods And Evaluation Criteria:**

The proposed method, NCAL, is well-aligned with the problem of data-driven personalized learning, particularly in addressing the long-tail distribution challenge in real-world benchmarks. The evaluation is conducted on two benchmark datasets. Extensive experiments demonstrate its effectiveness in improving model performance and mitigating class imbalance, making the evaluation criteria appropriate for the problem at hand.

**Other Comments Or Suggestions:**

Please see other reviews

**Other Strengths And Weaknesses:**

## Strengths
- The introduction of Textmodality Collapse (TC) regularization is a unique contribution that optimizes the distribution of text embeddings within the LLM representation space, significantly enhancing performance.

- NCAL is model-agnostic, meaning it can be integrated with various architectures and methods, making it highly adaptable for different use cases and ensuring wide practical applicability.


## Weaknesses
- Personalized learning is a crucial research topic in intelligent education. However, due to various constraints in real-world environments, the collected data often exhibit severe long-tail distributions. The authors propose the NCAL method, which leverages the concept of neural collapse and introduces TC regularization to constrain the model, mitigating the impact of long-tail distributions. Would it be possible to include additional visualizations, rather than relying solely on numerical metrics in Tables, to more intuitively demonstrate the effectiveness of the proposed method in enhancing category representation space?

- This paper uses the TIMSS and PMTD datasets to evaluate the effectiveness of this method. However, more information should be provided, such as the number of samples in each category, to give a more comprehensive view of the data distribution.

- Recently, some works [1] have used data augmentation to mitigate the impact of data-related issues. It is recommended to include experiments comparing these methods with the approach proposed in this paper.

[1] Towards Accurate and Fair Cognitive Diagnosis via Monotonic Data Augmentation

**Questions For Authors:**

I believe the authors have addressed an important and interesting problem. I am particularly concerned about whether the proposed method can be better visualized to clearly demonstrate how it addresses the long-tail problem in the representations. If this can be done, I would be willing to increase my score.

**Relation To Broader Scientific Literature:**

This paper builds upon prior research in neural collapse and representation learning by introducing NCAL, which optimizes feature representations through the simplex ETF structure and enhances text embedding distribution with TC regularization. However, the authors should improve the writing to enhance the clarity of the paper.

**Theoretical Claims:**

I have verified the theoretical claims in the paper, and the proofs are correct and well-founded.

---

> ### Author Rebuttal · Authors · 2025-04-01
>
> > **Q1: However, the assertion that NCAL is model-agnostic would benefit from further clarification.**
>
> **A1:** NCAL is designed to learn features that conform to the same simplex equiangular tight frame (ETF) structure by introducing TC regularization to optimize text distribution. As a result, NCAL serves as a plug-and-play framework. To further demonstrate its versatility, we present experimental results in Table 2, showcasing NCAL’s performance when integrated with different LLM architectures, such as Qwen2.5 and Llama3.1.
>
> > **Q2: More detailed examples of its impact on real-world class imbalance problems would provide further credibility to the claims.**
>
> **A2:** Thank you for your valuable suggestion. We visualized the class imbalance statistics of the personalized learning data collected from real-world scenarios used in this paper (see Q5). We also conducted experiments on other open-source datasets to evaluate NCAL’s performance under different class imbalance datasets. These results further demonstrate its effectiveness in practical applications (see Q3).
>
> > **Q3: However, incorporating additional personalized learning baselines or datasets.**
>
> **A3:** As suggested by the reviewer, we add a comprehensive study comparing four LLM methods and three open-source datasets, including: [IITJEE NEET AIIMS Students Questions Data](https://www.kaggle.com/datasets/mrutyunjaybiswal/iitjee-neet-aims-students-questions-data); [MathDial](https://github.com/eth-nlped/mathdial/tree/main); [DialogID](https://github.com/ai4ed/DialogID/tree/main). These results demonstrate the effectiveness of NCAL in comparison with more baselines under various long-tail personalized learning datasets. These results are shown below:
>
> | Model                       | Parameters | Dataset   | Acc       |
> | --------------------------- | ---------- | --------- | --------- |
> | DeepSeek-R1-Distill-Qwen-7B | 7B         | TMWPL     | 60.86     |
> | phi-4                       | 14B        | TMWPL     | 72.00     |
> | DeepSeek-V2 | 16B | TMWPL | 65.24 |
> | Qwen2.5-Instruct | 7B  | TMWPL | 61.14 |
> | **NCAL-Qwen2.5-Instruct**   | 7B    | TMWPL | **74.86** |
>
> | Model                 | Parameters | INASQD-Acc  | MathDial-Acc   | DialogID-Acc |
> | --------------------- | ---------- | -------- | ----- | --------------------- |
> | Qwen2.5-Instruct      | 7B         | 94.50 | 51.50        | 85.15 |
> | **NCAL-Qwen2.5-Instruct** | 7B       | **96.50** | **57.00** | **87.46** |
> | Llama3.1-Instruct | 8B | 95.50      | 51.00 | 85.75 |
> | **NCAL-Llama3.1-Instruct** | 8B       | **98.00** | **53.50** | **87.34** |
> | Qwen2.5-Math-Instruct | 7B        | 95.00 | 52.00 | 84.53 |
> | **NCAL-Qwen2.5-Math-Instruct** | 7B | **96.50** | **57.50** | **86.03** |
>
> > **Q4: Would it be possible to include additional visualizations?**
>
> **A4:** In Figure 1, we present the visualization to demonstrate the effectiveness of our method in improving the class feature space. The arrows indicate the direction of class centers, while the points represent sample features, with colors corresponding to the classes. In Figures 1 (a) and (c), it is visually evident that the baseline suffers from severe class representation imbalance. In (b) and (d), our method more uniformly learns the information for each class.
>
> > **Q5: However, more information should be provided, such as the number of samples in each category, to give a more comprehensive view of the data distribution.**
>
> **A5:** We thank the reviewer for pointing out this issue. We will include the statistical data in the final version. The data distribution is as follows, and it is clear that the number of samples in different categories exhibits a distinct long-tail distribution.
>
> | Classes  | Recall | Formulate | Identify | Represent | Implement | Inferences | Analyze |
> | ------- | ------ | --------- | -------- | --------- | --------- | ---------- | ------- |
> | TMWPL samples | 5045   | 1283      | 717      | 711       | 405       | 235        | 216     |
>
> | Classes  | R-FR | I-Q  | R-SR | F-I  | F-F  | U    | I-H  | R-RR |
> | ------- | ---- | ---- | ---- | ---- | ---- | ---- | ---- | ---- |
> | PMTD samples | 1014 | 880  | 737  | 518  | 510  | 389  | 104  | 62   |
>
> > **Q6: It is recommended to include experiments comparing these methods with the approach proposed in this paper.**
>
> **A6:** We agree with the reviewer that data augmentation is indeed an effective approach for addressing long-tail distributions. However, data augmentation requires strict control over the augmentation strategy, as well as the quality and diversity of the generated samples. In contrast, NCAL has broader applicability. It is important to note that we sincerely cannot provide a baseline comparison with the paper ``Towards Accurate and Fair Cognitive Diagnosis via Monotonic Data Augmentation,” as the augmentation method used in that work is based on ID-type data (e.g., 0, 1, 2), which cannot be learned using LLM encoding.

---

> > ### Comment · Reviewer_FPDE · 2025-04-06
> >
> > Thank you for the detailed response. After reviewing your rebuttal along with the comments from other reviewers, I believe most of the key concerns have been satisfactorily addressed. I am therefore increasing my score to reflect an acceptance recommendation

---

### Decision · Program_Chairs · 2025-05-01

**Decision:**

Accept (poster)

**Comment:**

This paper explores the use of neural collapse regularization on text embeddings during the fine-tuning of large language models (LLMs) for personalized learning. Experimental results demonstrate that the proposed approach improves performance across tasks and helps mitigate class imbalance.

The technical contribution of this paper seems limited, as the idea of promoting neural collapse to address long-tailed and imbalanced scenarios has been explored before. But this paper presents a strong application of the neural collapse principle to LLMs, particularly in data-driven personalized learning. In the revision, the authors are advised to incorporate the discussions with reviewers (such as the additional experiments) and add more discussions about other approaches in promoting neural collapse for imbalanced learning.